# WHAT WOULD THE EXPERT $do(\cdot)$?: CAUSAL IMITATION LEARNING

## ABSTRACT

We develop algorithms for imitation learning from policy data that was corrupted by unobserved confounders. Sources of such confounding include *(a)* persistent perturbations to actions or *(b)* the expert responding to a part of the state that the learner does not have access to. When a confounder affects multiple timesteps of recorded data, it can manifest as spurious correlations between states and actions that a learner might latch on to, leading to poor policy performance. To break up these spurious correlations, we apply modern variants of the classical *instrumental variable regression* (IVR) technique, enabling us to recover the causally correct underlying policy *without* requiring access to an interactive expert. In particular, we present two techniques, one of a generative-modeling flavor (`DoubIL`) that can utilize access to a simulator and one of a game-theoretic flavor (`ResiduIL`) that can be run entirely offline. We discuss, from the perspective of performance, the types of confounding under which it is better to use an IVR-based technique instead of behavioral cloning and vice versa. We find both of our algorithms compare favorably to behavioral cloning on a simulated rocket landing task.

## 1 INTRODUCTION

Much of the theory of imitation learning (IL) indicates that with enough demonstrations, we should be able to accurately recover the expert's policy. When we apply IL algorithms in practice however, we sometimes see them produce manifestly incorrect estimates of the expert's policy (Muller et al., 2006; Codevilla et al., 2019; de Haan et al., 2019; Bansal et al., 2018; Kuefler et al., 2017). One possible reason for this phenomenon is that empirically, we only have access to *noisy* recordings of what the expert did. This critical detail has been thus far neglected by most prior theoretical work in imitation learning. We focus in this paper on how best to learn from two kinds of noisy data:

- *Exogenous noise*: When we observe expert actions corrupted by a persistent noise (e.g. a faulty joystick that persistently perturbs actions before they are executed in the game).

- *Endogenous noise*: When we do not observe the full state an expert used to pick an action (e.g. the learner not knowing there's an enemy behind a door).

The net effect of either kind of persistent noise (more formally, an *unobserved confounder*) is to introduce temporal correlations in the recorded actions that do not have their true cause in the recorded state. For example, consider recordings of an expert driver slowing down at a stop sign. If all we present the learner with as state input is whether the expert was slowing down at the last timestep, they will likely learn to simply repeat the expert's past action. Thus, once the car begins to slow down, it continues to slow down, *regardless of whether there is a stop sign present*. At a more abstract level, these sorts of *inertia problems* can result from temporal correlations between pairs of actions (e.g. the effect of the stop sign) being reflected in the state (e.g. the past action variable), leading to spurious correlations between state and action that the learner might unfortunately latch onto (e.g. repeating the past action).

What should we hope to learn then in these confounded settings? Given we do not have access to the unobserved confounder, a reasonable choice is to ensure that we match the behavior of an expert that has access to the same information we do. That is, if we could *query* the expert for an action with only the information we have available, we should strive to produce an action that matches

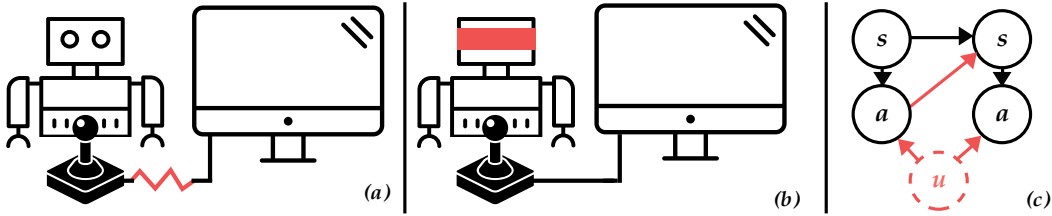

Figure 1: We focus in imitation learning in the presence of temporally correlated perturbations (exogenous noise, *(a)*) or not having access to the full state (endogenous noise, *(b)*). We formalize both in a graphical model *(c)* that allows us to leverage a technique known as *instrumental variable regression* to find a policy that isn't corrupted by spurious correlations introduced by the confounder.

this queried action. While applying an interactive imitation learning algorithm (Ross et al., 2011) would allow us to collect a dataset uncorrupted by confounding, a queryable expert is not a realistic assumption for many domains. We therefore focus on approaches for the off-policy setting. We base our algorithms on a technique from econometrics for dealing with confounding in recorded data known an *instrumental variable regression* (IVR) (Angrist et al., 1996). The high-level idea of IVR is to leverage an *instrument*, a source of random variation independent of the confounder, to deconfound inputs to a learning procedure via conditioning on the instrument. In dynamical systems, history can act as this source of variation, as it is unaffected by future confounding (Hefny et al., 2015). Our key insight is that *we can leverage past states as instruments to break the spurious correlation between states and actions caused by an unobserved confounder*.

Our work provides the following contributions:

**1. We formalize confounding in imitation learning.** We provide a structural causal model that captures inertia effects that result from temporally correlated exogenous or endogenous noise. We also derive a test to detect whether this sort of confounding is present in a dataset.

**2. We present a unified derivation of modern instrumental variable regression techniques.** We show how two recent extensions of the classical IVR technique share a common structure. We also extend the theoretical analysis of previous work by deriving accuracy bounds.

**3. We provide two novel algorithms to deal with confounding in imitation learning.** We derive two novel IVR-based algorithms:

- `DoubIL` is a generative modeling approach that and can utilize access to a simulator for reduced sample complexity.
- `ResiduIL` is a simulator-free, game-theoretic approach.

We derive performance bounds for policies produced by these algorithms under exogenous noise. We empirically investigate the effect of the persistence of the confounder on this bound. We also compare the performance of these approaches to behavioral cloning under endogenous noise.

## 2 RELATED WORK

**Imitation Learning.** Broadly speaking, imitation learning approaches can be grouped into three classes: offline, online, and interactive. Our work is most similar to offline imitation learning algorithms (e.g. Behavioral Cloning (Pomerleau, 1989), ValueDice (Kostrikov et al., 2019), AdVIL (Swamy et al., 2021)) that operate purely on collected data. Unlike previous work however, we consider the effect of unobserved confounding. Our work shares the goal of interactive imitation learning algorithms (e.g. DAgger (Ross et al., 2011), AggreVaTe (Ross & Bagnell, 2014)), in that we seek to match the output of a query to the expert. However, we focus on matching the output of a query on recorded data, rather than on *learner* rollouts, as is standard for interactive approaches. This is because the confounder decouples the actions recorded in the data and queried actions. Zhang et al. (2020) consider imitation learning through the lens of causal inference but focus on the one-step setting, while we consider multiple timesteps. Kumor et al. (2021) contemporaneously consider the

| PROBLEM | CORRELATED STATE | ACTION | CONFOUNDER |
|---|---|---|---|
| Gridlock | Distance to intersection | Crossing | Cars on other side of intersection (*endog.*) |
| Stationary | Previously Braking | Braking | Traffic light (*endog.*) |
| Faulty Brakes | Speed | Braking | Brakes not responding to presses (*exog.*) |

Table 1: Concrete examples of confounding from the driving domain. The stationary problem was observed empirically by Codevilla et al. (2019).

multi-step setting and come to similar conclusions as us about the challenges of endogenous noise. They derive a necessary and sufficient structural condition for successful imitation learning, while we focus on practical algorithms with performance guarantees for a particular graphical model under exogenous noise. In contrast to imitation learning methods that seeks to match moments of the expert's behavior (Swamy et al., 2021), we focus only on matching average expert actions. We leave matching arbitrary moments to future work.

**Inertia Effects in Imitation Learning.** Several authors have empirically observed a latching effect in policies trained via imitation learning: (Muller et al., 2006; Codevilla et al., 2019; de Haan et al., 2019; Bansal et al., 2018; Kuefler et al., 2017), where learned policies tend to inappropriately repeat the same action. We seek to provide a plausible explanation and correction for the phenomenon reported in these works, and list several examples in Table 1. We note that when attempting to explain inertia effects, de Haan et al. (2019) propose causal confounding as the root cause of the error. However, as previously pointed out by Spencer et al. (2021), there is no actual confound in the theoretical or empirical examples in the work of de Haan et al. (2019). This is because the learner observes all of the variables the expert was using to make decisions.

**Instrumental Variable Regression.** The classical approach to instrumental variable regression (Wright, 1928) is a two-stage least squares procedure (e.g. in Angrist et al. (1996)'s textbook). We focus on the more general nonlinear setting and instead base our approaches on the more recent DEEPIV (Hartford et al., 2017) and AGMM (Dikkala et al., 2020). We present extensions to the work in these papers, including a unified derivation of both methods and error analysis for DEEPIV.

## 3 A Brief Review of Instruments in Causal Modeling

We begin by discussing the concept of an instrument before deriving our algorithmic approaches in a simplified, non-sequential setting. Let $X$, $Y$, and $Z$ be random variables on (potentially infinite) sample spaces $\mathcal{X}$, $\mathcal{Y}$, and $\mathcal{Z}$. Assume that $X$, $Y$, and $Z$ have the causal, rather than statistical, dependency structure in Fig. 2. Given a dataset of $(x, y, z)$ tuples, we are interested in determining the causal relationship between $X$ and $Y$, $\mathbb{E}[Y|do(x)]$, where $do(\cdot)$ is the interventional operator of Pearl et al. (2016). Intuitively, $\mathbb{E}[Y|do(x)]$ is the expected value of $Y$ when we *intervene* and set $X = x$, rather than observe such an $X$. In the SCM to the right, $h(x) = \mathbb{E}[Y|do(x)]$. Because of the presence of an unobserved confounder, $U$, that affects both $X$ and $Y$, standard regression (e.g. Ordinary Least Squares or OLS) generically produces inconsistent estimates. Coarsely, this occurs because OLS will over-estimate the influence of the parts of $X$ that are affected by the confounder. If we only have observational data and are unable to perform randomized control trials, a canonical technique to recover $h$ is IVR (Winship & Morgan, 1999). Formally, an *instrument* $Z$ must satisfy three structural conditions:

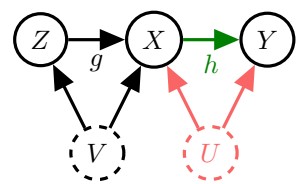

Figure 2: The structural causal model (SCM) we consider. We are interested in finding $h$, the causal relationship from $X$ to $Y$, even though there is an unobserved confounder, $U$. To do so, we leverage the effect of $Z$, which provides independent randomness from $U$.

1. *Unconfounded Instrument*: $Z \perp\!\!\!\perp U$ – i.e. independent randomization from confounder.
2. *Exclusion*: $Z \perp\!\!\!\perp Y|X, U$ – i.e. no extraneous paths.
3. *Relevance*: $Z \not\perp\!\!\!\perp X$ – i.e. conditioning has an effect.

$Z$ satisfies these three conditions in the SCM of Fig. 2. [1] Without loss of generality, we assume that $\mathbb{E}[U] = 0$. This allows us to concisely derive a set of *conditional moment restrictions* (CMR):

$$0 = \mathbb{E}[U] = \mathbb{E}[U|z] = \mathbb{E}[Y - h(X)|z] \tag{1}$$

$$\Rightarrow \forall z \in \mathcal{Z}, \ \mathbb{E}[Y|z] = \mathbb{E}[h(X)|z]. \tag{2}$$

In words, these constraints are saying that a necessary condition for recovery of $h(x)$ is that for all values of the instrument, the actual and predicted expected values of $Y|Z$ are equal. We further assume that noise $U$ enters additively to $Y$,[2] and write out the following equations:

$$X = g(Z, U, V), \quad Y = h(X) + U. \tag{3}$$

We now derive an appropriate loss function for finding an $\widehat{h}$ that approximately satisfies the CMR. If we only have finite samples and can therefore only estimate conditional expectations up to some tolerance, it is natural to relax the CMR to

$$
\begin{aligned}
\min_{\widehat{h} \in \mathcal{H}, \boldsymbol{\delta}} \quad & \tfrac{1}{2}\mathbb{E}_z[\delta_z^2] \\
\text{s.t.} \quad & |\mathbb{E}[Y - \widehat{h}(X)|z]| \leq \delta_z, \quad \delta_z \geq 0, \quad \forall z \in \mathcal{Z},
\end{aligned} \tag{4}
$$

where the $\delta_z$ are slack variables. Then, the Lagrangian (with the natural $P(z)$-weighted inner product that captures how often each we expect each $z$ to occur) is

$$L(\widehat{h}, \boldsymbol{\delta}, \boldsymbol{\lambda}) = \sum_{z \in \mathcal{Z}} P(z)\lambda_z(\mathbb{E}[Y - \widehat{h}(X)|z] - \delta_z) + P(z)\frac{1}{2}\delta_z^2, \tag{5}$$

where $\boldsymbol{\lambda}$ is the vector of Lagrange multipliers. By the stationarity component of the KKT conditions,

$$\nabla_{\delta_z} L(\widehat{h}, \boldsymbol{\delta}, \boldsymbol{\lambda}) = -P(z)\lambda_z + P(z)\delta_z = 0, \tag{6}$$

implying that $\delta_z = \lambda_z$. Plugging this back into the Lagrangian, we can simplify our function to

$$L(\widehat{h}, \boldsymbol{\lambda}) = \sum_{z \in \mathcal{Z}} P(z)\lambda_z \mathbb{E}[Y - \widehat{h}(X)|z] - P(z)\frac{1}{2}\lambda_z^2. \tag{7}$$

We refer to (7) as the *Regularized Lagrangian* or ReLa for short. Now, solving for the optimal Lagrange multipliers via stationarity, we arrive at

$$\nabla_{\lambda_z} L(\widehat{h}, \boldsymbol{\lambda}) = P(z)\mathbb{E}[Y - \widehat{h}(X)|z] - P(z)\lambda_z = 0, \tag{8}$$

which implies the optimal $\lambda_z$ is equal to $\mathbb{E}[Y - \widehat{h}(X)|z]$. Plugging this back into (7) recovers the loss function,

$$L(\widehat{h}) = \sum_{z \in \mathcal{Z}} P(z)\mathbb{E}[Y - \widehat{h}(X)|z]^2 = \text{PRMSE}^2(\widehat{h}). \tag{9}$$

This expression is the square of the *Projected Root Mean Squared Error* (PRMSE) of Chen & Pouzo (2012). To recap, by minimizing Eq. 9, we are attempting to find an $\widehat{h}$ that approximately satisfies the CMR. Minimizing PRMSE is a necessary condition for recovering $\mathbb{E}[Y|do(X)]$. For it to be a sufficient condition, one needs the natural identifiability assumptions – we refer interested readers to Chen & Pouzo (2012) for a more thorough discussion.

## 3.1 GENERATIVE MODELING APPROACH

How should we minimize the PRMSE then? One option is learning the distribution $P(X|z) = g(z)$, passing samples from it to a candidate $\widehat{h}$, and trying to match $\mathbb{E}[Y|z]$. This is a generalization of the standard Two-Stage Least Squares (2SLS) (Angrist et al., 1996) procedure to nonlinear functions. The nonlinearity of the second stage means that one cannot simply compute the first moment of the $P(X|z)$ distribution, which is recovered by linearly regressing from $X$ to $Z$ in the 2SLS procedure. This sort of approach was first proposed for the IVR setting by Hartford et al. (2017) and amounts to first learning a $g(z)$ (e.g. via maximum likelihood estimation) and then solving

$$\min_{\widehat{h} \in \mathcal{H}} \mathbb{E}_Z \left[ (\mathbb{E}[Y|z] - \mathbb{E}_{\hat{X} \sim g(z)}[\widehat{h}(\hat{X})])^2 \right]. \tag{10}$$

The work of Hartford et al. (2017) did not have theoretical analysis regarding the effect of errors in $g(z)$ upon attempts to learn $h(x)$. We prove the following in Appendix A:

---

[1]The inclusion of $V$ makes our model a generalization of the standard IVR model, so we confirm the validity of the instrument in Appendix A.

[2]Without this assumption, one can only upper/lower bound $h(x)$ (Balke & Pearl, 2013).

**Theorem 1.** *Assume we learn a $g(z)$ s.t. $\max_{\widehat{h} \in \mathcal{H}} \mathbb{E}_Z[(\mathbb{E}_{x \sim g(z)}[\widehat{h}(x)] - \mathbb{E}_{x \sim P(X|z)}[\widehat{h}(x)])^2] \leq \delta$. Then, optimizing (10) to value $\epsilon$ corresponds to recovering a $\widehat{h}(x)$ s.t. PRMSE$(\widehat{h}) \leq \sqrt{\delta} + \sqrt{\epsilon}$.*

### 3.2 GAME-THEORETIC APPROACH

One can also proceed by instead solving the two-player zero-sum game with the ReLa (7) as the payoff. Denoting by $f \in \mathcal{F} = \{\mathcal{Z} \to \mathbb{R}\}$ the function that maps $z$'s to their Lagrange multipliers, we can write this game as

$$\min_{\widehat{h} \in \mathcal{H}} \max_{f \in \mathcal{F}} \mathbb{E}[2(Y - \widehat{h}(X))f(Z) - f(Z)^2]. \tag{11}$$

This game is the core objective of the AGMM method of Dikkala et al. (2020). Importantly, one does not need to learn a generative model of $P(X|z)$ for these sorts of game-theoretic approaches. We prove the following theorem in Appendix A:

**Theorem 2.** *Assume that $\mathcal{H}$ and $\mathcal{F}$ are bounded, closed under negation, convex, compact, and that $h \in \mathcal{H}$ and $\forall \widehat{h} \in \mathcal{H}$, $f(z) = \mathbb{E}[Y - \widehat{h}(X)|z] \in \mathcal{F}$. Then, an $\epsilon$-approximate Nash equilibrium of (11) corresponds to recovering a $\widehat{h}(x)$ s.t. PRMSE$(\widehat{h}) \leq \sqrt{\epsilon}$.*

One can find such an equilibrium via a standard reduction to no-regret online learning (Freund & Schapire, 1997).

In summary, one can frame nonlinear IVR as a generative modeling or game-theoretic problem, leading to different error characteristics. We now turn our attention to applying these methods to imitation learning with unobserved confounders.

## 4 CAUSAL CONFOUNDING IN IMITATION LEARNING

We begin with a brief, intuitive sketch to illustrate the challenges of confounding for IL: consider an expert trying to fly a quadcopter straight but their actions being perturbed by wind (i.e. a form of exogenous noise). Because it attempts to reproduce expert actions, behavioral cloning would reproduce these deviations, producing trajectories that deviate *even further* from a straight path in a windy environment. In contrast, by filtering out the effects of the confounder, a policy trained via IVR would only be affected by the wind present at test time and therefore produce trajectories similar to those of the expert.

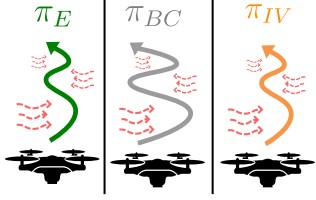

We now formalize this sort of confounding and how one can use IVR to mitigate its effects. We use $\Delta(S)$ to mean the set of distributions over $S$ and focus on a Markov Decision Process (MDP) parameterized by $\langle \mathcal{S}, \mathcal{A}, \mathcal{T}, r, T \rangle$, where $\mathcal{S}$ is the state space, $\mathcal{A}$

Figure 3: Behavioral cloning amplifies the effect of exogenous noise, unlike IV.

is the action space, $\mathcal{T} : \mathcal{S} \times \mathcal{A} \to \Delta(\mathcal{S})$ is the transition operator, $r : \mathcal{S} \times \mathcal{A} \to [-1, 1]$ is the reward function, and $T$ is the horizon of the problem. Let $J(\pi) = \mathbb{E}_{\tau \sim \pi}[\sum_{t=1}^{T} r(s_t, a_t)]$, $\Pi \subseteq \{\mathcal{S} \to \Delta(\mathcal{A})\}$ be the policy class we optimize over and $d_\pi$ be the visitation distribution of policy $\pi$. In the presence of unobserved confounding, the trajectories generated by the expert can be captured by the structural causal model (SCM) in Fig. 4. [3] Fig. 4 captures both the exogenous and endogenous noise settings. In the exogenous noise setting, the confounder $u_{t-1}$ could be a persistent noise that affects pairs of actions, while in the endogenous noise setting, $u_{t-1}$ can be thought of as a response to a part of the state that the learner does not have access to (see Table 1 for several example) In either setting, the confounding travels through the dynamics to influence the next state, leading to spurious correlations between the recorded states and actions. We can also see this correlative effect by writing out the structural equations:

$$X = s_t = \mathcal{T}(s_{t-1}, a_{t-1}) = \mathcal{T}(s_{t-1}, \pi_E(s_{t-1}) + u_{t-1} + u_{t-2}) \tag{12}$$

$$Y = a_t = \pi_E(s_t) + u_t + u_{t-1}. \tag{13}$$

---

[3]One technically needs to add another input to $\pi_E$ (i.e. $\pi_E(s, R)$, where $R$ is a random input) to allow for a non-deterministic expert. In this work, we focus on techniques for minimizing the PRMSE, for which matching average expert actions is sufficient. Thus, we suppress the dependence on the other input.

Fig. 4 also tells us that $Z = s_{t-1}$ satisfies the three conditions to make it a valid instrument for countering the effects of $U = u_{t-1}$. Intuitively, this is because the past state is independent of the current confounder, allowing it to function as an independent source of randomness. One can imagine longer time-scale correlations induced between actions than just one step confounding – our approaches naturally extend to this setting by using a state further back in the past as the instrument.

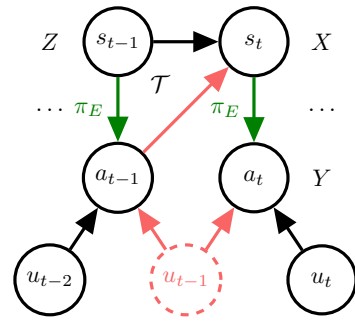

Unlike standard imitation learning approaches like behavior cloning which attempt to recover $\mathbb{E}[a|s]$, an approach based on IVR enables us to instead recover the interventional effect of the policy $\mathbb{E}[\pi_E(s)|s] = \mathbb{E}[a|do(s)]$. Conceptually, $\mathbb{E}[a|do(s)]$ is asking what the expert would do on average if they were placed in state $s$, *i.e.* the kind of answer we would get from a queryable expert in DAgger (Ross et al., 2011) or DAeQuIL (Swamy et al., 2021). However, as we are only interested in the result of queries on states from expert demonstrations, we are able to get a similar effect via IVR to an interactive approach without requiring access to a queryable expert. We now build upon this intuition to derive two algorithms.

Figure 4: The SCM for imitation learning with unobserved confounders. The confounding is mediated via the dynamics into the state, introducing spurious correlations between states ($X = s_t$) and actions ($Y = a_t$). To break the confounding, we can utilize the past state as an instrument ($Z = s_{t-1}$).

## 5 ALGORITHMS FOR CAUSAL IMITATION LEARNING

We now present two approaches for causal imitation learning that can be seen as applications of the generative modeling and game-theoretic approaches of Sec. 3. At their core, both algorithms are attempting to minimize a PRMSE objective,

$$\min_{\pi \in \Pi} \mathbb{E}_{(s,s',a') \sim d_{\pi_E}} [(\mathbb{E}[a' - \pi(s')|s])^2], \tag{14}$$

instead of the usual IL objective,

$$\min_{\pi \in \Pi} \mathbb{E}_{(s,a) \sim d_{\pi_E}} [(a - \pi(s))^2]. \tag{15}$$

In both the exogenous and endogenous settings, minimizing equation 14 corresponds to recovering $\mathbb{E}[a|do(s)]$. What differs is the performance implications of doing so, as we now discuss further.

### 5.1 EXOGENOUS NOISE

Exogenous noise is present both in the demonstrations as well as at test time. Our goal in this setting is to eliminate the effect of the confounder so at test time we do not needlessly reproduce its effects (e.g. the increased swerving in our quadcopter example). Notice that under exogenous noise, minimizing (15) to 0 would not recover the expert's policy while minimizing Equation (14) would. First, let a distribution $P(U)$ be c-Total Variation stable (Bassily et al., 2021) if:

$$\|a - b\|_2 \leq \delta \Rightarrow d_{TV}(a + U, b + U) \leq c\delta. \tag{16}$$

This property is satisfied by a wide variety of distributions. For example, for standard normal random variables, $c = 1/2$. Next, in our setting, the measure of ill-posedness (Dikkala et al., 2020; Chen & Pouzo, 2012) is

$$\kappa(\Pi) = \sup_{\pi \in \Pi} \frac{\sqrt{\mathbb{E}_{s \sim d_{\pi_E}} [(\pi_E(s) - \pi(s))^2]}}{\sqrt{\mathbb{E}_{s,s',a' \sim d_{\pi_E}} [\mathbb{E}[a' - \pi(s')|s]]^2}} = \sup_{\pi \in \Pi} \frac{\text{RMSE}(\pi)}{\text{PRMSE}(\pi)}. \tag{17}$$

We prove the following bound on policy performance in Appendix B:

**Theorem 3.** *Assume $P(U)$ is c-TV Stable exog. noise, $\pi_E$ is deterministic, and let $\kappa(\Pi)$ be the measure of the ill-posedness of the problem. Then, PRMSE$(\pi) \leq \epsilon \Rightarrow J(\pi_E) - J(\pi) \leq c\kappa(\Pi)\epsilon T^2$.*

Intuitively, $\kappa(\Pi)$ measures the strength of the strength of the instrument. Consider the extreme case where $s' = s$. Then, $\kappa(\Pi) = 1$. As the past state becomes a weaker instrument, $\kappa(\Pi) > 1$. Thus, if the confounding affects multiple timesteps, we would expect $\kappa(\Pi)$ to grow as one needs to reach further back in time to find a valid instrument, leading to a looser performance bound. We investigate the effect of the length of confounding on the ill-posedness of the problem empirically in Sec. 6.

## 5.2 ENDOGENOUS NOISE

Endogenous noise is present *only* in the expert demonstrations and not applied during learner roll-outs. Thus, in contrast to the exogenous setting, we do not need to eliminate the effect of the confounder to perform as well as the expert. Instead, we hope to effectively reduce our uncertainty over the confounder (e.g. the state of the traffic light). We begin by defining the following policies:

$$\forall s \in \text{supp}(d_{\pi_E}), \quad \pi_{BC}(a|s) = d_{\pi_E}(s,a)/d_{\pi_E}(s), \quad \pi_{IV}(a|s) = p(a|do(s)), \quad (18)$$

where supp denotes support. We prove the following results under endog. noise in Appendix A:

**Lemma 1.** *There exist MDPs for which $\pi_E$, $\pi_{BC}$, and $\pi_{IV}$ have different trajectory distributions.*

**Lemma 2.** *If reward $r$ is a function of state and action only, $J(\pi_E) = J(\pi_{BC})$ always, while there exist MDPs for which $J(\pi_E) > J(\pi_{IV})$.*

**Lemma 3.** *If reward $r \in \{\mathcal{S} \times \mathcal{A} \times \mathcal{U} \to \mathbb{R}\}$ (i.e. the reward additionally depends on the confounder), then there exist MDPs for which $J(\pi_E) > J(\pi) \, \forall \pi \in \{\mathcal{S} \to \Delta(\mathcal{A})\}$.*

One take-away from these lemmas is the fundamental difficulty of producing a value equivalent policy to the expert's under endogenous noise, a result that was concurrently derived via a graphical condition by Kumor et al. (2021) (Defn. 2.3). Consider, for example, an expert driver that stops at an intersection when a traffic light is red. If the learner does not see this light, there is no way for them to ensure they match the behavior of such an expert. However, in the special case where the reward function does not directly depend on the confounder (i.e. eliminating the natural reward function that penalizes the learner for not obeying the traffic light), $\pi_{BC}$ is value equivalent to the expert, while, perhaps counter-intuitively, causally consistent $\pi_{IV}$ is not. The value difference is because $\pi_{IV}$ marginalizes out $u_{t-1}$ by sampling from $P(u_{t-1})$ while $\pi_{BC}$ samples from $P(u_{t-1}|s_t)$ (see proof of Lemma 1 in Appendix A). $\pi_{BC}$ is therefore better able to estimate the value of the confounder by utilizing the information in the current state, which is advantageous in the endogenous setting.

## 5.3 WITH A SIMULATOR: DoubIL

---

**Algorithm 1** DoubIL

---

**Input:** Dataset $\mathcal{D}_E$ of expert trajectories, Policy class $\Pi$, Simulator $\widehat{\mathcal{T}}$
**Output:** Trained policy $\pi_2$
$\pi_1 = \arg\min_{\pi \in \Pi} \mathbb{E}_{s,a \sim \mathcal{D}_E}[-\log \pi(a|s)]$      {Train preliminary policy via moment-matching.}
$\mathcal{D}_{IV} = \{(\widehat{\mathcal{T}}(s, \pi_1(s)), a') | \forall (s, a') \in \mathcal{D}_E\}$      {Pass $\pi_1$'s actions through simulator.}
$\pi_2 = \arg\min_{\pi \in \Pi} \mathbb{E}_{s,a \sim \mathcal{D}_{IV}}[(a - \pi(s))^2]$      {Train final policy on new dataset and output.}

---

Algorithm 1 can be seen as a variation of generative modeling approach of Sec. 3 and Hartford et al. (2017) where one leverages knowledge of one factor of the $P(X|z)$ distribution and just learns the other factor. Via the Markov assumption, we can factorize $P(X|z) = P(S'|s) = \sum_{a \in \mathcal{A}} P(a|s)\mathcal{T}(s,a)$. Assuming access to a simulator $\widehat{\mathcal{T}}$ that closely

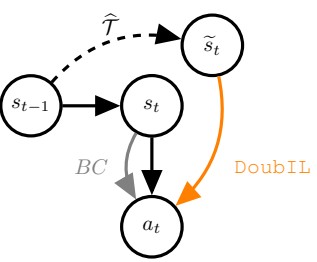

approximates the true transition dynamics, we can instead focus on learning the $P(a|s)$ component: the standard imitation learning task. Notably, this first-stage policy is biased as it includes the effect of the confounder: $P(a|s) = P(U + \pi_E(s)|s)$. However, when we use it to simulate transitions, the next states that are produced no longer have the

Figure 5: DoubIL deconfounds inputs to the second stage by re-simulating state transitions.

particular instantiation of the confounder present in the recorded dataset's next actions. Using a tilde to denote a fresh draw from a distribution, simulated states are drawn from

$$\widetilde{s}_t \sim \widehat{\mathcal{T}}(s_{t-1}, \pi_1(s_{t-1})) \tag{19}$$

while the observed next actions are drawn from

$$a_t \sim \pi_E(\mathcal{T}(s_{t-1}, \pi_E(s_{t-1}) + u_{t-1} + u_{t-2})) + u_{t-1} + u_t. \tag{20}$$

Notice that there are no shared noise terms. This allows us to apply standard imitation learning to this new dataset of $(\widetilde{s}_t, a_t)$ to learn a causally consistent policy. The two applications of imitation learning lead us to term this algorithm `DoubIL`. We can translate the guarantee of Theorem 1 to our factored context:

**Lemma 4.** *Assume we learn a $\pi_1(s)$ s.t.*

$$\max_{\pi \in \Pi} \mathbb{E}_{s_{t-1}} \left[ (\mathbb{E}_{s_t \sim \widehat{\mathcal{T}}(s_{t-1}, \pi_1(s_{t-1}))}[\pi(s_t)] - \mathbb{E}_{s_t \sim P(s_t | s_{t-1})}[\pi(s_t)])^2 \right] \leq \delta$$

*Then, optimizing the second-stage MSE to $\epsilon$ corresponds to recovering a $\pi_2$ s.t.*

$$PRMSE(\pi_2) = \sqrt{\mathbb{E}_{s \sim d_{\pi_E}}[\mathbb{E}[\pi_2(s') - \pi_E(s') | s]^2]} \leq \sqrt{\delta} + \sqrt{\epsilon}.$$

We prove this lemma in Appendix A. Combining this lemma with Theorem 3 allows one to derive a performance bound of $J(\pi_E) - J(\pi) \leq c\kappa(\Pi)(\sqrt{\delta} + \sqrt{\epsilon})T^2$ under exogenous noise. We note that one could simply learn the mapping $P(s'|s)$ but this can be far less sample efficient than merely learning a policy when $|\mathcal{A}| \leq |\mathcal{S}|$, as is often true in practice.

### 5.4 WITHOUT STATE RE-SAMPLING: RESIDUIL

---

**Algorithm 2** `ResiduIL`

---

**Input:** Dataset $\mathcal{D}_E$ of expert trajectories, Policy class $\Pi$, Discriminator class $\mathcal{F}$, Learning rate $\eta$
**Output:** Trained policy $\pi$
Set $\pi \in \Pi, f \in \mathcal{F}, \widetilde{g}_\pi = 0, \widetilde{g}_f = 0$
**while** $\pi$ not satisfactory **do**
    $L(\pi, f) = \mathbb{E}_{(s,s',a') \sim \mathcal{D}_E}[2(a' - \pi(s'))f(s) - f(s)^2]$         {Payoff of zero-sum game.}
    $g_\pi = \nabla_\pi L(\pi, f), g_f = \nabla_f L(\pi, f)$         {Perform Optimistic Mirror Descent.}
    $\pi \leftarrow \pi - \eta(2g_\pi - \widetilde{g}_\pi)$
    $f \leftarrow f + \eta(2g_f - \widetilde{g}_f)$
    $\widetilde{g}_\pi \leftarrow g_\pi, \widetilde{g}_f \leftarrow g_f$
**end while**

---

Algorithm 2 is the direct application of the game-theoretic approach of Sec. 3 and Dikkala et al. (2020) to imitation learning. We term it `ResiduIL` because the adversary attempts to predict the residual between the learner and the expert's actions while the learner attempts to minimize this residual. Notably, this algorithm can be run completely offline (i.e. without access to a simulator). We use the Optimistic Mirror Descent approach of Syrgkanis et al. (2015) to find approximate Nash equilibria in our experiments. Once again, we can extend our past results to the IL setting:

**Lemma 5.** *An $\epsilon$-approximate equilibrium for the policy player corresponds to recovering a policy $\pi$ s.t $PRMSE(\pi) \leq \sqrt{\epsilon}$.*

This lemma dovetails with Theorem 3 to prove that $J(\pi_E) - J(\pi) \leq c\kappa(\Pi)\sqrt{\epsilon}T^2$ under exogenous noise (Appendix A).

## 6 EXPERIMENTS

We test `DoubIL` and `ResiduIL` on a slightly modified version of the OpenAI Gym (Brockman et al., 2016) LunarLander-v2 environment against a behavioral cloning baseline. We generate

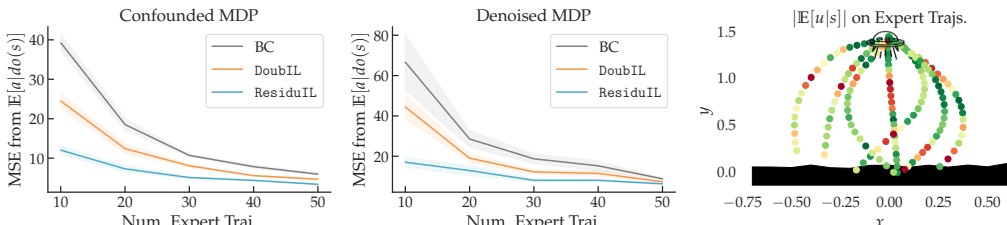

Figure 6: We train behavioral cloning, `DoubIL`, and `ResiduIL` on trajectories from a modified LunarLander environment, computing standard errors across 5 runs. The left plot shows that `DoubIL` and `ResiduIL` are better able to match the desired $\mathbb{E}[a|do(s)]$ on states from expert rollouts, while the middle plot shows they are able to generalize better to the state dist. of an expert w/o noise. The right plot shows how we can compare the results of behavioral cloning and causal IL procedures to identify areas of the state space where the effect of confounding is strong (the red dots).

demonstrations by simulating rollouts of an expert policy trained via PPO (Schulman et al., 2017), adding fresh Gaussian noise to the expert's action as well as cached noise from the last timestep. The latter noise is the confounder. See Appendix B for full parameters. We judge policy quality by computing the MSE between the output of a deconfounded expert query and a learner's proposed actions on states from expert rollouts. We see that both of our methods are able to more closely match $\mathbb{E}[a|do(s)]$ than behavioral cloning, especially in the low-data regime (Fig. 6, left). We also measure the MSE on states from deconfounded expert rollouts – while there are no clear guarantees on this state distribution, we see that our methods generalize better than BC empirically (Fig. 6, middle). One might wonder how, given a dataset of expert demonstrations, one detects whether there is unobserved confounding in the data. We can answer this question by comparing the results of behavioral cloning and either of our above algorithms. We prove the following in Appendix A:

**Lemma 6.** *Assume $\pi_{BC}(s) = \mathbb{E}[a|s]$ and $\pi_{IV}(s) = \mathbb{E}[a|do(s)]$. Then, $\mathbb{E}[u|s] = \pi_{BC}(s) - \pi_{IV}(s)$.*

The implication of this lemma is that comparing the outputs of IVR-based procedures to behavioral cloning can help us detect causal confounding – if they greatly differ with a sufficiently sized dataset, there is likely an unobserved confounding effect in our data. Moreover, the states where they differ represent the parts of the state space where the influence of the confounder is highest. Fig. 6 right is an empirical example of how the test of Lemma 6 can be used to identify areas of the state space where the effect of the confounder is especially strong (e.g. the center).

For linear problems, we can bound $\kappa(\Pi)$ (the measure of ill-posedness) via an eigenvalue ratio (Dikkala et al., 2020). Extending our previous model to include the effect of the last $H$ confounders ($a_t = \pi_E(s_t) + \sum_{j=t-H}^{t} u_j$.), we arrive at the bound

$$\kappa(\Pi; H) \leq \sqrt{\frac{\lambda_{max}(\mathbb{E}[s_t s_t^T])}{\lambda_{min}(\mathbb{E}[\mathbb{E}[s_t|s_{t-H}]\mathbb{E}[s_t|s_{t-H}]^T])}}. \qquad (21)$$

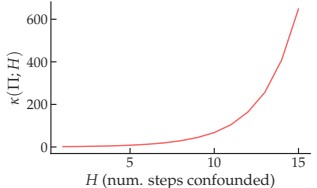

We compute this quantity empirically for a linear-quadratic problem with Gaussian confounding and plot results in Fig. 7. As expected, we see that increasing the length of confounding leads to weaker instruments as one has to use states further back in time. Theorem 3 tells us that under exogenous noise, we should expect a weaker instrument to lead to a larger performance gap between the learner and expert. See Appendix B for full experimental setup details.

Figure 7: We compute $\kappa(\Pi)$ for an LQG problem where we vary the number of steps a confounder sticks around for.

## 7 CONCLUSION

We present a model that captures confounding in imitation learning and derive two algorithms, `DoubIL` and `ResiduIL`, that are able to utilize history as an instrument to mitigate the effects of unobserved confounders. We prove performance bounds and validate their empirical efficacy under exogenous noise. We also discuss the challenges of learning with endogenous noise.

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

# A PROOFS

## A.1 PROOF OF VALIDITY OF INSTRUMENT

*Proof.* We check the instrument conditions in order:

1. *Unconfounded Instrument*: $Z \perp\!\!\!\perp U$: The $Z \to X \leftarrow U$, $V \to X \leftarrow U$, and $X \to Y \leftarrow U$ triples are blocked by standard d-separation rules (Pearl et al., 2016). All paths from $Z$ to $U$ must pass through one of these triples so $Z \perp\!\!\!\perp U$.

2. *Exclusion*: $Z \perp\!\!\!\perp Y | X, U$: The $Z \to X \to Y$, $X \leftarrow U \to Y$, and $V \to X \to Y$ triples are blocked by standard d-separation rules. All paths from $Z$ to $Y$ must pass through one of these triples so $Z \perp\!\!\!\perp Y | X, U$.

3. *Relevance*: $Z \not\perp\!\!\!\perp X$: There is a $Z \to X$ edge, which is assumed to be non-degenerate.

Thus, $Z$ is a valid instrument for determining the causal relationship between $X$ and $Y$. $\qquad\square$

## A.2 PROOF OF THEOREM 1

*Proof.* We simplify notation for clarity in our proof. Consider two vectors of the same dimension, $\mathbf{a}$ and $\mathbf{b}$. Assume that $\sum_i^N a_i^2 \leq \epsilon$ and $\sum_i^N b_i^2 \leq \delta$. This implies that $\|\mathbf{a}\|_2 \leq \sqrt{\epsilon}$ and $\|\mathbf{b}\|_2 \leq \sqrt{\delta}$. Then, by the triangle inequality, $\|\mathbf{a} - \mathbf{b}\|_2 \leq \|\mathbf{a}\|_2 + \|\mathbf{b}\|_2 \leq \sqrt{\epsilon} + \sqrt{\delta}$. Setting $a_i = \sqrt{P(z)}(\mathbb{E}[Y|z] - \mathbb{E}_{\hat{x} \sim g(z)}[\hat{h}(\hat{x})])$ and $b_i = \sqrt{P(z)}(\mathbb{E}_{\hat{x} \sim g(z)}[\hat{h}(\hat{x})] - \mathbb{E}[\hat{h}(x)|z])$ proves that

$$\max_{\hat{h} \in \mathcal{H}} \mathbb{E}_Z[(\mathbb{E}_{x \sim g(z)}[\hat{h}(x)] - \mathbb{E}_{x \sim P(X|z)}[\hat{h}(x)])^2] \leq \delta, \tag{22}$$

$$\mathbb{E}_z[(\mathbb{E}[Y|z] - \mathbb{E}_{\hat{x} \sim g(z)}[\hat{h}(\hat{x})])^2] \leq \epsilon \tag{23}$$

$$\Rightarrow PRMSE(\hat{h}) = \sqrt{\mathbb{E}_z[(\mathbb{E}[Y|z] - \mathbb{E}_{x \sim P(X|z)}[\hat{h}(x)])^2]} \leq \sqrt{\epsilon} + \sqrt{\delta} \tag{24}$$

$$\square$$

## A.3 PROOF OF THEOREM 2

*Proof.* The population version of (11) is

$$\min_{h \in \mathcal{H}} \max_{f \in \mathcal{F}} \mathbb{E}[2(Y - h(X))f(Z) - f^2(Z)] \tag{25}$$

An $\epsilon$-approximate equilibrium is an $(\hat{h}, \hat{f})$ pair such that:

$$\max_{f \in \mathcal{F}} \mathbb{E}[2(Y - \hat{h}(X))f(Z) - f^2(Z)] - \frac{\epsilon}{2} \tag{26}$$

$$\leq \mathbb{E}[2(Y - \hat{f}(X))\hat{f}(Z) - \hat{f}^2(Z)] \tag{27}$$

$$\leq \min_{h \in \mathcal{H}} \mathbb{E}[2(Y - h(X))\hat{f}(Z) - \hat{f}^2(Z)] + \frac{\epsilon}{2} \tag{28}$$

Taking the derivative w.r.t $f(z)$ of the payoff and setting it equal to 0, we arrive at

$$2P(z)\mathbb{E}[Y - \hat{h}(X)|z] - 2P(z)f(z) = 0 \Rightarrow f(z) = \mathbb{E}[Y - \hat{h}(X)|z]. \tag{29}$$

Plugging this back into (49) gives us the inequality

$$\mathbb{E}_Z[\mathbb{E}[Y - \hat{h}(X)|z]^2] - \frac{\epsilon}{2} \leq \min_{h \in \mathcal{H}} \mathbb{E}[2(Y - h(X))\hat{f}(Z) - \hat{f}^2(Z)] + \frac{\epsilon}{2}. \tag{30}$$

Assuming we are in the realizable setting (e.g. $h(x) = \mathbb{E}[Y|do(x)] \in \mathcal{H}$), $\min_{h \in \mathcal{H}} \mathbb{E}[2(Y - h(X))\hat{f}(Z) - \hat{f}^2(Z)] \leq 0$. Thus, we can write that:

$$\mathbb{E}_Z[\mathbb{E}[Y - \hat{h}(X)|z]^2] - \frac{\epsilon}{2} \leq \frac{\epsilon}{2} \Rightarrow PRMSE(\hat{h}) \leq \sqrt{\epsilon}. \tag{31}$$

$$\square$$

We note that Theorem 2 follows somewhat directly from the main theorems of Dikkala et al. (2020) but that it was not stated in this precise form in their work.

### A.4 Proof of Lemma 1

*Proof.* We focus on the first two timesteps of the problem. By construction, $p(s_0)$ is the same for all three policies. The following statements follow from standard conditional independence rules (Russell & Norvig, 2002):

$$\pi_{BC}(a_0|s_0) = p(a_0|s_0) = \sum_{u_0} P(u_0)P(a_0|s_0, u_0), \tag{32}$$

$$\pi_{IV}(a_0|s_0) = p(a_0|do(s_0)) = \sum_{u_0} P(u_0)P(a_0|do(s_0), u_0) = \sum_{u_0} P(u_0)P(a_0|s_0, u_0). \tag{33}$$

The last equality follows from the fact $s_0$ has no parents. Thus, when combined with the fact that the transition dynamics $P(s_1|s_0, a_0)$ are the same for all three policies, we arrive at the following equality:

$$p_{\pi_E}(s_0, a_0, s_1) = p_{\pi_{BC}}(s_0, a_0, s_1) = p_{\pi_{IV}}(s_0, a_0, s_1) \tag{34}$$

The first difference between the trajectory distributions starts at $a_1$. By definition, the expert chooses actions via

$$\pi_E(a_1|s_1, u_0, u_1) = p(a_1|s_1, u_0, u_1). \tag{35}$$

while the learners instead follow

$$\pi_{BC}(a_1|s_1) = p(a_1|s_1) = \sum_{u_0} \sum_{u_1} p(u_0|s_1)p(u_1)p(a_1|s_1, u_0, u_1), \tag{36}$$

$$\pi_{IV}(a_1|s_1) = p(a_1|do(s_1)) = \sum_{u_0} \sum_{u_1} p(u_0)p(u_1)p(a_1|do(s_1), u_0, u_1) \tag{37}$$

$$= \sum_{u_0} \sum_{u_1} p(u_0)p(u_1)p(a_1|s_1, u_0, u_1). \tag{38}$$

The first line follows from the fact $u_1 \perp\!\!\!\perp u_0, s_1$. The last equality follows from Rule 2 of do-calculus (Pearl et al., 2016). Thus, given the action distributions at the second step are not equal, so long as $a_1 \not\perp\!\!\!\perp u_0$ and $s_1 \not\perp\!\!\!\perp u_0$, all three policies have different trajectory distributions.

$\square$

### A.5 Proof of Lemma 2

*Proof.* We prove $J(\pi_E) = J(\pi_{BC})$ via an application of the Performance Difference Lemma (PDL) (Kakade & Langford, 2002). We note that this is equivalent to the graphical argument used by Kumor et al. (2021) (Defn. 2.3 in their paper).

First, we note that $p(s_0, a_0)$ is equal for both policies. Assuming that $p(s_{t-1}, a_{t-1})$ is equal, $p(s_t)$ must be equal by the fact the transition dynamics are the same. Then, via the fact that $\pi_{BC}$ matches the conditional distribution of $a_t|s_t$, $p(s_t, a_t)$ must also be equal. Thus, by induction, $d_{\pi_{BC}} = d_{\pi_E}$. This means that $\pi_{BC}$ is defined properly everywhere within it's own state visitation distribution. We then apply the PDL as follows:

$$J(\pi_E) - J(\pi_{BC}) = T\mathbb{E}_{s,a,u\sim d_\pi}[Q^{\pi_E}(s, a) - \mathbb{E}_{a'\sim p(a'|s,u)}[Q^{\pi_E}(s, a')]] \tag{39}$$

$$= T\mathbb{E}_{s,a\sim d_\pi}[Q^{\pi_E}(s, a) - \mathbb{E}_u[\mathbb{E}_{a'\sim p(a'|s,u)}[Q^{\pi_E}(s, a')]]] \tag{40}$$

$$= T\mathbb{E}_{s,a\sim d_\pi}[Q^{\pi_E}(s, a) - \mathbb{E}_{u|s}[\mathbb{E}_{a'\sim p(a'|s,u)}[Q^{\pi_E}(s, a')]]] \tag{41}$$

$$= T\mathbb{E}_{s,a\sim d_\pi}[Q^{\pi_E}(s, a) - \mathbb{E}_{a'\sim p(a'|s)}[Q^{\pi_E}(s, a')]] \tag{42}$$

$$= 0. \tag{43}$$

The third equality follows from the fact the learner's actions are independent of the confounder, so under $d_{\pi_{BC}}$, $P(u|s) = P(u)$. From the proof of the previous lemma, $p(a_1|s_1) \neq p(a_1|do(s_1))$ in general, so $\pi_{IV}$ and $\pi_E$ can have different state-action visitation distributions after the first timestep. Therefore, for a two-step problem where $a_1 \not\perp\!\!\!\perp u_0$ and $s_1 \not\perp\!\!\!\perp u_0$,

$$J(\pi_E) - J(\pi_{IV}) = T(\mathbb{E}_{s,a\sim d_{\pi_E}}[r(s, a)] - \mathbb{E}_{s,a\sim d_{\pi_{IV}}}[r(s, a)]) > 0 \tag{44}$$

for reward function

$$r(s, a) = \begin{cases} 1, d_{\pi_E}(s, a) > d_{\pi_{IV}}(s, a) \\ 0, o.w. \end{cases}. \tag{45}$$

$\square$

A.6 PROOF OF LEMMA 3

*Proof.* Let $u_t$ be a Rademacher random variable (1 w.p. $\frac{1}{2}$ and $-1$ otherwise) $\forall t \in [T]$. Let $a_t | s_t, u_t, u_{t-1} = u_t$ and there be a single fixed state $s$ that no action can leave. Set $r(s, a_t, u_t) = \mathbf{1}[a_t = u_t]$. Then, $J(\pi_E) = T$. Notice that $p(+1|s) = p(-1|s) = p(+1|do(s)) = p(-1|do(s)) = 0.5$. Thus, $J(\pi_E) > J(\pi_{IV}) = J(\pi_{BC}) = \frac{T}{2}$, the expected number of correct answers of randomly guessing the outcome of a fair coin $T$ times. Furthermore, notice that $\frac{T}{2}$ is the best any $\pi \in \Pi$ can do. One can see this by considering the problem with $T = 1$. If there was a policy $\pi$ with $J(\pi) > \frac{1}{2}$, one would be able to use such a policy to predict the outcome of a fair coin better than the Bayes-optimal classifier for coin-betting ($p(\text{heads}) = p(\text{tails}) = \frac{1}{2}$), violating the fact the Bayes-optimal classifier minimizes Bayes error. □

A.7 PROOF OF LEMMA 4

*Proof.* Notice that

$$\max_{\pi \in \Pi} \mathbb{E}_{s_{t-1}}[(\mathbb{E}_{s_t \sim \widehat{\mathcal{T}}(s_{t-1}, \pi_1(s_{t-1}))}[\pi(s_t)] - \mathbb{E}_{s_t \sim P(s_t|s_{t-1})}[\pi(s_t)])^2] \leq \delta \tag{46}$$

can be re-written as

$$\max_{\pi \in \Pi} \mathbb{E}_Z[(\mathbb{E}_{x \sim g(z)}[\pi(x)] - \mathbb{E}_{x \sim P(X|z)}[\pi(x)])^2] \leq \delta. \tag{47}$$

Thus, the proof of Theorem 4 holds as written. □

A.8 PROOF OF LEMMA 5

An $\epsilon$-approximate equilibrium for the policy player is a $\pi$ such that

$$\max_{f \in \mathcal{F}} \mathbb{E}[2(a_t - \pi(s_t))f(s_{t-1}) - f^2(s_{t-1})] - \frac{\epsilon}{2} \leq \min_{\pi \in \Pi} \mathbb{E}[2(a_t - h(s_t))\widehat{f}(s_{t-1}) - \widehat{f}^2(s_{t-1})] + \frac{\epsilon}{2}. \tag{48}$$

With a change of notation, we can re-write this as:

$$\max_{f \in \mathcal{F}} \mathbb{E}[2(Y - \pi(X))f(Z) - f^2(Z)] - \frac{\epsilon}{2} \leq \min_{\pi \in \Pi} \mathbb{E}[2(Y - h(X))\widehat{f}(Z) - \widehat{f}^2(Z)] + \frac{\epsilon}{2}. \tag{49}$$

Thus, the proof of Theorem 2 holds as written.

A.9 PROOF OF THEOREM 3

*Proof.* By definition,

$$\text{PRMSE}(\pi) = \sqrt{\mathbb{E}_{s \sim d_{\pi_E}}[\mathbb{E}[a' - \pi(s')|s]]^2} = \epsilon. \tag{50}$$

Recall that the measure of ill-posedness of the problem (Dikkala et al., 2020; Chen & Pouzo, 2012) can be defined as

$$\kappa(\Pi) = \sup_{\pi \in \Pi} \frac{\sqrt{\mathbb{E}_{s \sim d_{\pi_E}}[(\pi_E(s) - \pi(s))^2]}}{\sqrt{\mathbb{E}_{s, s', a' \sim d_{\pi_E}}[\mathbb{E}[a' - \pi(s')|s]]^2}} = \sup_{\pi \in \Pi} \frac{\text{RMSE}(\pi)}{\text{PRMSE}(\pi)} \tag{51}$$

Directly,

$$\text{RMSE}(\pi) \leq \epsilon \kappa(\Pi) \tag{52}$$

We repeat the definition of total variation stability of a distribution $P(U)$:

$$\|a - b\|_2 \leq \delta \Rightarrow d_{TV}(a + U, b + U) \leq c\delta. \tag{53}$$

We proceed by noting that TV-stability implies that $\forall s \in \mathcal{S}$,

$$d_{TV}(\pi(s) + U, \pi_E(s) + U) \leq c\|\pi(s) - \pi_E(s)\| \tag{54}$$

$$\Rightarrow d_{TV}(\pi(s) + U, \pi_E(s) + U)^2 \leq c^2\|\pi(s) - \pi_E(s)\|^2 \tag{55}$$

$$\Rightarrow \mathbb{E}_{s \sim d_{\pi_E}}[d_{TV}(\pi(s) + U, \pi_E(s) + U)^2] \leq c^2 \mathbb{E}_{s \sim d_{\pi_E}}[\|\pi(s) - \pi_E(s)\|^2] = c^2 \text{MSE}(\pi). \quad (56)$$

By Jensen's inequality,

$$\mathbb{E}_{s \sim d_{\pi_E}}[d_{TV}(\pi(s) + U, \pi_E(s) + U)]^2 \leq \mathbb{E}_{s \sim d_{\pi_E}}[d_{TV}(\pi(s) + U, \pi_E(s) + U)^2] \leq c^2 \text{MSE}(\pi). \quad (57)$$

Taking the square root of both sides, we arrive at

$$\mathbb{E}_{s \sim d_{\pi_E}}[d_{TV}(\pi(s) + U, \pi_E(s) + U)] \leq c \, \text{RMSE}(\pi) \leq c\kappa(\Pi)\epsilon. \quad (58)$$

Lastly, we apply the Performance Difference Lemma of Kakade & Langford (2002) as follows:

$$J(\pi_E) - J(\pi) = T\mathbb{E}_{s,a \sim d_{\pi_E}}[Q^\pi(s,a) - \mathbb{E}_{a' \sim \pi(s)}[Q^\pi(s,a')]] \quad (59)$$

$$= T\mathbb{E}_{s,a \sim d_{\pi_E}}[Q^\pi(s, \pi_E(s) + u + \widetilde{u}_1) - \mathbb{E}[Q^\pi(s, \pi(s) + u + \widetilde{u}_2)]] \quad (60)$$

$$\leq T^2 \mathbb{E}_{s \sim d_{\pi_E}}[d_{TV}(\pi(s) + U, \pi_E(s) + U)] \quad (61)$$

$$\leq c\kappa(\Pi)\epsilon T^2. \quad (62)$$

We use the fact that the same $u$ would be added to both the learner and the expert's actions and that rewards are in the range $[-1, 1]$ in the third step.

$\square$

## A.10 PROOF OF LEMMA 6

*Proof.*

$$\mathbb{E}[a_t | do(s_t)] = \mathbb{E}[\pi_E(s_t) + u_t + u_{t-1} | do(s_t)] = \pi_E(s_t) + \mathbb{E}[u_t] + \mathbb{E}[u_{t-1}] = \pi_E(s_t) \quad (63)$$

$$\mathbb{E}[a_t | s_t] = \mathbb{E}[\pi_E(s_t) + u_t + u_{t-1} | s_t] = \pi_E(s_t) + \mathbb{E}[u_t] + \mathbb{E}[u_{t-1} | s_t] = \pi_E(s) + \mathbb{E}[u_{t-1} | s_t] \quad (64)$$

$$\pi_{BC}(s) - \pi_E(s) = \mathbb{E}[a_t | s_t] - \mathbb{E}[a_t | do(s_t)] = \mathbb{E}[u_{t-1} | s_t] = \mathbb{E}[u | s] \quad (65)$$

$\square$

# B EXPERIMENT DETAILS

## B.1 LUNARLANDER EXPERIMENTS

For ease of simulation, we remove the legs from the LunarLander vehicle (the joints connecting them to the main body have a state that is not recorded in the observed state), remove the dispersion noise, and generate trajectories with a fixed ground layout.

For all learned functions, we use two-layer ReLu MLPs with 64 hidden units. We use the Adam optimizer (Kingma & Ba, 2014) for behavioral cloning and `DoubIL` and use the optimistic variant for `ResiduIL`. We apply a weight decay of 1e-3 to all. We train all methods for 50k steps.

| PARAMETER | VALUE |
|---|---|
| LEARNING RATE | 3E-4 |
| BATCH SIZE | 128 |

Table 2: Parameters for behavioral cloning.

For computational ease, we only learn the mean of $P(a|s)$ for `DoubIL` and add fresh standard normal noise on-top of it to simulate drawing actions. For more complex noise models, one would need to use a moment matching algorithm (Swamy et al., 2021) in the first stage.

Importantly, `DoubIL` suffers from a "double-sample" issue (Baird, 1995) where multiple independent samples of $g(z)$ are required to compute gradients of $\widehat{h}$. To see this, note that the gradient with respect to $h$ of (10) is

$$\mathbb{E}_Z \left[ (\mathbb{E}[Y|z] - \mathbb{E}_{\hat{x} \sim \hat{g}(z)}[\widehat{h}(\hat{x})])(-\mathbb{E}_{\hat{x} \sim \hat{g}(z)}[\frac{\partial}{\partial \widehat{h}} \widehat{h}(\hat{x})]) \right] \quad (66)$$

| PARAMETER | VALUE |
|---|---|
| LEARNING RATE | 3E-4 |
| BATCH SIZE | 128 |
| NUM. SAMPLES FOR $\mathbb{E}$ | 8 |

Table 3: Parameters for `DoubIL`.

Notice that $\hat{x}$ appears under two *separate* expectations that are then multiplied together. To get an unbiased estimate of this product, two samples of $\hat{x}$ are required, one for each expectation. Therefore, it is most correct to use multiple samples from $g(z)$ for each update.

Thus, for implementing the "double samples" for the gradient, we compute $\mathbb{E}_1[a' - \pi(s')|s]$ and $\mathbb{E}_2[a' - \pi(s')|s]$ using independent samples. Then, we apply a stop-gradient operator to the former expectation before taking a product between the expectations and averaging over $s$:

$$L(\pi) = \mathbb{E}_s[\oslash(\mathbb{E}_1[a' - \pi(s')|s])\mathbb{E}_2[a' - \pi(s')|s]]. \tag{67}$$

This loss function has the correct gradient as it uses independent samples for computing the two expectations.

| PARAMETER | VALUE |
|---|---|
| LEARNING RATE | 5E-5 |
| BATCH SIZE | 128 |
| BC REGULARIZER WEIGHT | 5E-2 |
| $f$ NORM PENALTY | 1E-3 |
| ADAM $\beta$S | 0, 1E-2 |

Table 4: Parameters for `ResiduIL`.

## B.2 LQG EXPERIMENTS

We compute the optimal policy for the following canonical linear system via solving a Discrete-Time Algebraic Ricatti Equation via the standard iterative method:

$$x_t = Ax_{t-1} + Bu_{t-1} \tag{68}$$

$$J(K) = \sum_t^T x_t^T Q x_t + (Kx_t)^T RK x_t \tag{69}$$

$$A = \begin{bmatrix} 1 & \Delta T \\ 0 & 1 \end{bmatrix}, B = \begin{bmatrix} 0.5(\Delta T)^2 \\ \Delta T \end{bmatrix}, Q = \begin{bmatrix} 1 & 0 \\ 0 & 1 \end{bmatrix}, R = [0.1], \Delta T = 0.1$$

This is the dynamics of a "sliding brick on a frozen lake." We then simulate rollouts of 200 timesteps with $u_t$ being drawn i.i.d. from the standard normal distribution. We confound actions with the sum of confounders going $H$ steps back:

$$a_t = K^* s_t + \sum_{j=t-H}^t u_j. \tag{70}$$

We simulate 1000 such rollouts to compute (21) empirically. We calculate $\mathbb{E}[X|z] = \mathbb{E}[s_t|s_{t-H}] = (A + BK^*)^H s_{t-H}$ analytically instead of via samples due to the small value of the quantity in comparison to the variance of the noise.

