# OpenReview forum: "What Would the Expert $do(\cdot)$?: Causal Imitation Learning"
_ICLR.cc/2022/Conference — ICLR 2022 Submitted_

### Official Review · Reviewer_A5Fo · 2021-10-19

**Correctness:** 3
**Technical Novelty And Significance:** 3
**Empirical Novelty And Significance:** 3
**Recommendation:** 6
**Confidence:** 3

**Main Review:**

Strengths:

•	One of the first papers to consider causal imitation learning

•	Error bounds are provided for two known IVR approaches

•	Experiments show the benefit of explicitly accounting for hidden confounders

Weaknesses:

•	Can the author provide some examples and justifications of the following two assumptions made by the proposed method (Fig 4)?

States and confounders are marginally independent. Take the authors’ quadcopter as an example, the positions (which I assume these are the states) are most likely not independent of the wind unless the expert can perfectly fly it straight?

Confounders are independent of each other. Using the same example, the wind at this time step is probably not independent of the wind of the last time step? If the confounders have temporal dependence, it seems to introduce additional dependence between states and confounders via action (e.g., considering the path u[t], u[t-1], u[t-2], a[t-1], s[t])

•	Experiments. This is related to the first weakness. Right now, the experiment is set up so that the assumptions are perfectly satisfied by design. What if the assumptions are violated to different degrees? How robust is the proposed method?


**Summary Of The Paper:**

This paper proposes two causal imitation learning algorithms to address the confounding issue that causes spurious correlations. The method relies on instrumental variable regression to correct for confounding effects. The main idea is to use the state variable from the last time point as instrumental variable for the state-act variables at the current time point in a Markov decision process.

**Summary Of The Review:**

Overall, I find the paper to be novel and interesting.  It could be improved, in my opinion, in terms of the suitability/justification of the IVR assumptions in the considered application domain as I mentioned in the weaknesses above.

---

> ### Author Response · Authors · 2021-11-12
> **Re:**
>
> We thank the reviewer for their comments and appreciate that they recognized the novelty of our work. Responding to the concerns raised in the weakness section:
> 1. “States and confounders are marginally independent.”  We don’t believe we are making this assumption -- $P(u_{t-1}|s_t)$ is not equal to $P(u_{t-1})$. One can view our model as perhaps the simplest possible generalization of the standard PGM for an MDP -- we simply allow noise to be temporally correlated, instead of the usual iid noise assumption.
> 2. “Confounders are independent of each other.” This is an excellent point we were thinking of adding more discussion around. If we merge all correlated confounders into a single node U, one can see that a valid instrument is then a state that is not affected by this set of noises. There must always exist such a state if the initial state distribution is not confounded. As we detail in the last experiment of the paper, this means that one needs to optimize PRMSE to a much lower value to derive a non-vacuous performance bound.
> 3. Experiments: We agree our paper would benefit from more experiments showing the robustness to violated assumptions (e.g. having the confounder effect multiple timesteps, but only using the immediately prior state as an instrument). We will try to add such experiments to an updated draft.

---

> > ### Comment · Reviewer_A5Fo · 2021-11-17
> > **#1**
> >
> > Thank you for your response. Regarding the 1st response, how about u[t-1] and s[t-1]? They seem to be marginally independent in Figure 4?

---

> > > ### Author Response · Authors · 2021-11-21
> > > **Re:**
> > >
> > > $u_{t-1}$ and $s_{t-1}$ are indeed independent. We chose this graphical model as a simple generalization of i.i.d. noise -- it is an interesting question to consider more complex noise models with additional dependencies.

---

### Official Review · Reviewer_Jzke · 2021-11-02

**Correctness:** 2
**Technical Novelty And Significance:** 2
**Empirical Novelty And Significance:** 2
**Recommendation:** 3
**Confidence:** 3

**Main Review:**

Overall, I think the authors consider an exciting problem, since imitation learning has been widely applied in reinforcement learning and the presence of unobserved confounders is always prevalent in practical applications. However, it does not seem that the problem formulation of this paper properly addresses the challenges of causal imitation learning. More importantly, cloning the causal effect of the state S on action A does not necessarily allow the imitator to achieve the expert's performance. For instance, consider a causal model where S and A are both determined by an unobserved variable U in {0, 1}. That is, S <- U and A <-U, P(U = 1) = 0.5. The reward Y <- A \xor \neg S. In this case, the interventional distribution P(A = 1 | do(S = 1)) = P(A = 1 | do(S = 0)) = 0.5, i.e., do(S = s) has no causal effect on the expert's action A. If the imitator naively learns a policy copying P(A | do(S)), it is verifiable that it will achieve the expected reward equal to 0.5, which is far from the optimal expert's reward E[Y].

Indeed, the optimal imitation strategy here is to copy the observational distribution P(A|S), which leads to a policy f: A <- S. This allows the imitator to match the expert's reward. Such an imitation strategy is effective since S is backdoor-admissible w.r.t. A and Y. The completeness of the backdoor criterion in the single-stage causal imitation has been discussed in (Zhang et al., 2020) [27]. I understand the authors' goal to extend (Zhang et al., 2020) to the general sequential decision-making setting. However, it would be recommended if the authors could build on the existing framework of causal imitation learning. Finally, the backdoor condition has been extended to perform imitation learning in the sequential decision-making settings and is shown to be sufficient and necessary (Kumor, Zhang and Bareinboim, NeurIPS'21).

**Summary Of The Paper:**

This paper attempts to study imitation learning from a causal inference perspective. More specifically, let S be an observed state, A be action and Z be an instrumental variable. The authors propose that one should perform causal imitation learning by learning a policy function that could induce the same interventional distribution P(A|do(Z)).

**Summary Of The Review:**

This paper studies an interesting problem in imitation learning with the presence of unobserved confounders. The causal identification method with instrumental variables in Section 3 seems interesting. However, as demonstrated in the example in "Main Review", I am not sure if estimating the causal effect of observed state S on action A is sufficient for a successful imitation. In other words, I am not convinced that the proposed approach in this paper is technically sound.

---

> ### Author Response · Authors · 2021-11-12
> **Re:**
>
> We thank the review for their comments and share their excitement on this topic. Responding:
> 1. In our paper, we make a distinction between exogenous noise (present in training data and at test time) and endogenous noise (only present in the demonstrations). The excellent example the reviewer provides is an example of the latter, while the proposed algorithms (DoubIL and ResiduIL) are meant for the former. We are rather certain that under the additivity and zero-mean assumptions we make, they are technically sound and furthermore allow us to derive policy performance bounds.
> 2. We really appreciate the reference to the Kumor et al. paper -- we were not aware of it at the time of submission and it was a treat to read. We agree that the methods we propose in this paper would not be able to handle the endogenous setting well, and have cited the Kumor et al. paper for making this point in our updated draft. In some sense, we view our work as complementary to the work of Kumor et al.: while they have given conditions under which BC will produce consistent estimates of the expert policy, we have given algorithms that produce consistent policy estimates in (some) situations when BC does not.

---

> > ### Comment · Reviewer_Jzke · 2021-11-19
> > **Exogenous vs Endogenous**
> >
> > >'"In our paper, we make a distinction between exogenous noise (present in training data and at test time) and endogenous noise (only present in the demonstrations). The excellent example the reviewer provides is an example of the latter, while the proposed algorithms (DoubIL and ResiduIL) are meant for the former."
> >
> > I am a bit confused here. Do the authors imply that the unobserved variable U in my example is endogenous? Could the authors please elaborate on the "distinction between exogenous noise (present in training data and at test time) and endogenous noise (only present in the demonstrations)." I tried to find these definitions in the manuscript but failed to do so.

---

> > > ### Author Response · Authors · 2021-11-21
> > > **Re:**
> > >
> > > We are more than happy to do so:
> > > - We use *exogenous noise* to denote perturbations to actions (e.g. because of a faulty joystick) that are applied on top of a policy's chosen action. These are therefore present both at training time (i.e. in the expert demonstrations) as well as at test time (i.e. the joystick will still be faulty during learner rollouts). Regardless of whether the learner wants it or not, the temporally correlated noise perturbs their actions. In this setting, the goal of the learner should be to recover what the expert's action would have been without the noise. If they execute such a policy, it will match the trajectory distribution of the demonstrations as it will have exogenous noise applied on top of its proposed actions. Under an assumption of the noise entering additively, our two proposed approaches are capable of finding such a policy.
> > > - We use *endogenous noise* to refer to responses to parts of the state that the expert sees but the learner does not (e.g. a stop sign their sensors do not pick up). This kind of noise is therefore present at training time (in the sense we see its effects in the recorded data) but not at test time (because the learner still cannot see the stop sign). As in the example you give, if the reward function depends somehow on whether the stop sign is there, following the $P(a|do(s))$ policy does not give us value-equivalence to the expert. Intuitively, unless we were able to somehow infer the existence of a stop sign, we should not hope to match expert performance.
> > >
> > > We apologize for any confusion and have tried to add additional text around this point in the manuscript. Does it make more sense now?

---

> > > > ### Comment · Reviewer_Jzke · 2021-11-30
> > > > **Challenges of unobserved confounders**
> > > >
> > > > The authors claim that the proposed methods are not applicable if there exists a mismatch in the input states between the learner and the expert. That is, there is no unobserved confounder affecting the expert's action. However, it has been shown in (Zhang et al., 2020) that in such cases, standard imitation learning methods (i.e., behavior cloning and inverse RL) are ensured to identify the imitating policy. It is unclear how this paper improves over the existing RL literature despite the introduction of causal language. Therefore, I will keep my current score.

---

### Official Review · Reviewer_X9N9 · 2021-11-04

**Correctness:** 3
**Technical Novelty And Significance:** 2
**Empirical Novelty And Significance:** 2
**Recommendation:** 5
**Confidence:** 4

**Main Review:**

Strengths:
- Formulating latent confounders in imitation learning in the framework of structural causal models
- Proposing two algorithms to address the problem of imitation learning for the latent confounders considered in this work: In the first algorithm, if we have access to a simulator, the algorithm focuses on learning $P(a|s)$ by first drawing the next state from the simulator, and then try to learn a causally consistent policy. In the second algorithm, the authors utilized the approach in Dikkala et al. (2020).

Weaknesses:
- Some parts of the paper are not written rigorously and it is hard to follow these parts. For instance: 1- In Section 3.1, the authors argued that it is "most correct" to use multiple samples from $g(z)$. What do you mean by "most correct"? 2- At the beginning of Section 4, the authors mentioned some events that result in poor performance in behavior cloning. Again, this part cannot be easily justified. 3- On page 6, the exact modelling of exogenous and endogenous noises is not clear. What are the main differences between these two kinds of noises? 4- For the right plot in Figure 6, it seems that there is no explanation in the text. There are only some descriptions in the caption but it does not help to understand this plot.
- Issues regarding the models of latent confounders: Could the authors give a real-world example of latent confounders that can be modelled by Figure 4? It seems to me that additive noises are too restrictive in this model.
- About the first algorithm in Section 6, I did not understand how we can connect to instrumental variables and how the results in the previous section can be useful in this section.
- It would be great if the authors could explain how they removed the absolute function in eq. (4) when they wrote the Lagrangian in eq. (5).
- In the experiment section, it is better to evaluate the algorithms in more practical settings instead of adding some Gaussian noise.





**Summary Of The Paper:**

The authors proposed a SCM to model latent confounder in the problem of imitation learning. In particular, two cases of exogenous and endogenous noises are considered. For the instrumental variable, they studied the effect of error in estimation $P(X|z)=g(z)$ on Projected Root Mean Squared Error (PRMSE) where $Z$ is the instrumental variable. They also proposed a game-theoric approach to estimate $h(x)= \mathbb{E}[Y|do(X)=x]$. Moreover, they proposed two algorithms to deal with latent confounders in imitation learning. Experimental results showed the proposed algorithms can perform better with respect to behavior cloning.

**Summary Of The Review:**

In overall, some parts of the paper are not well-written. Moreover, the model of latent confounder (additive noise with special form in Figure 4) is not justified and the application of the proposed algorithms might be too restrictive.

---

> ### Author Response · Authors · 2021-11-12
> **Re:**
>
> We thank the reviewer for their comments. Responding in order to the concerns raised in the Weaknesses section:
> 1. We apologize for the imprecise wording. Using independent samples is required for unbiased estimates of the gradient. Without an unbiased gradient, even in the limit of infinite data, one has no easy guarantees on convergence to a minimum. However, in practice, one can for some problems get away without independent samples, as mentioned in the Hartford et al. paper. Thankfully, with access to a simulator, one can generate independent samples for our setting, so we use the correct approach in our experiments and DoubIL algorithm.
> 2. We presented the example at the beginning of Section 4 only to provide intuition for the sorts of negative effects confounding can have in imitation learning. We then show, both theoretically (Section 5) and empirically (Section 6), how this intuition captures a real issue.
> We agree that we should have given more discussion to the difference between endogenous and exogenous noise. In short, in the trajectories observed at training time, both fit into the same graphical model. At test time however, the endogenous noise no longer affects the process (because it was generated by the expert responding to something the learner does not see like a stop light) while exogenous still does (because it is something like wind that does not care about what policy is being run). We added more text around this point.
> 3. We attempted to provide some descriptions of the figure in the text (see the paragraph below Lemma 3). It would be helpful if the reviewer could point out what parts of this description were not sufficiently clear.
> 4. We appreciate the question about additivity. As we note in Footnote 2, without this assumption, one cannot in general identify the causal effect. This assumption is standard in the instrumental variable regression literature (see the Dikkala et al., Angrist et al. and Hartford et al. works). We further note that additive noise is an extremely common assumption, perhaps representing the majority of regression models used in the world.
> 5. Sorry, did you mean the first algorithm of Section 5? If so, the connection is that in the graphical model of figure 4, the past state can be used as an instrument, allowing us to utilize the IVR approaches of the past section to find the causal relationship between state (X) and action (Y).
> 6. We apologize for the concision in the derivation of PRMSE, we were pressed for space. We need an additional $\delta_z \geq 0$ constraint. Then, one can move from inequality constraint $|\mathbb{E}[Y-h(X)|Z]| \leq \delta_z$ to  $|\mathbb{E}[Y-h(X)|Z]| = \delta_z$ because of the objective function pushing down $\delta_z$. Lastly, we can recognize that removing both the absolute value and the $\delta_z \geq 0$ constraints will not change the feasible set and form the Lagrangian as usual.

---

> > ### Comment · Reviewer_X9N9 · 2021-11-28
> > **Thanks for addressing my comments**
> >
> > Thanks for addressing my comments. About the third comment, it is better to mention which one of u_i's are exogenous. About the fifth comment, I meant Algorithm 1. I decided to keep my score unchanged.

---

### Official Review · Reviewer_aRw4 · 2021-11-06

**Correctness:** 2
**Technical Novelty And Significance:** 2
**Empirical Novelty And Significance:** 2
**Recommendation:** 3
**Confidence:** 3

**Main Review:**

Strengths:

The authors have adapted the instrumental variable regression of Angrist etal. to estimate a better state transition for a linear dynamical system that has a Gaussian noise.

Weaknesses:

The paper describes a linear dynamical system expressed in causal language.  Fig 4 depicts a dynamical system that only has a similar form to the SCM in Angrist metal, but is not a SCM.  The DAG is not depicting the mechanism of data formation.  If the model does not account explicitly for all the causal factors [wind force, traveling medium (air density, air humidity, air temperature), copter state (position, velocity, acceleration, battery power), navigator actions, etc.], it is unclear that the Gaussian noise assumption is reasonable.  How was the noise distribution estimated?

By the authors' own admission they are not estimating the effects of interventions.

"We note that minimizing PRMSE does not guarantee recovery of E[Y|do(X)]. However, given that E[Y |do(X)] always minimizes PRMSE, a low PRMSE is indicative of being close to E[Y |do(X)]."

What is the definition of "close"?  "Close" by comparison to what? Given that the authors are not recovering E[Y|do(X)], the title of the paper "What would the Expert  do(.)?"  is misleading.

**Summary Of The Paper:**

The authors derive two algorithms based on instrumental variable regression and game theoretic approach for imitation learning.

**Summary Of The Review:**

The authors have adapted the instrumental variable work of Angrist etal. to estimate a better state transition for a linear dynamical system when the noise is Gaussian.  However, the authors do not seem to be modeling the mechanism of data formation, and they are not estimating the effects of interventions.  The Gaussian noise assumption has not been justified.

The authors are performing regression and the causal language is misleading.

---

> ### Author Response · Authors · 2021-11-12
> **Re:**
>
> We thank the reviewer for their comments. We would like to clarify a few points of potential confusion:
> 1. At a high level, our paper is focused on imitation learning: given trajectories of an expert acting in a dynamical system, can we extract the policy they were following? Unlike much of the past work in this space, we consider a setting where confounders ($u_t$) affect the demonstrated actions, while not being recorded in the data presented to the learner. In short, we are interested in estimating the policy that is acting in the dynamical system, not the state transition function / dynamics.
> 2. We are not exclusively focused on the Linear-Quadratic-Gaussian setting. For the last experiment in the paper, we chose an LQG problem because there is a closed-form expression for the ill-posedness of the problem, kappa. All of our analysis and our methods work outside of this setting (i.e. we do not assume linearity).
> 3. The main simplifying assumptions we make are of zero-mean noise and additive confounding. The former is common in most regression contexts (e.g. even for linear regression, if the noise had nonzero mean, you could only recover the function up to an additive shift). The latter is unique to the causal context -- as detailed in Footnote 2, without it, one cannot compute $E[Y|do(X)]$ from observational data. These assumptions are standard in the instrumental variable regression literature (see the Dikkala et al., Angrist et al. and Hartford et al. works).
> 4. Our experiments were not meant to realistically model the noise a quadcopter would face while flying. As noted by the reviewer, one has no a-priori reason to believe this is Gaussian. Rather, we were attempting to show that if our assumptions were satisfied, our proposed algorithms would be better able to remove the effect of the confounder than standard regression (behavioral cloning in the imitation learning setting).
> 5. The DAG we consider (figure 4) would be able to account for the factors the reviewer mentions (the causal factors as part of the dynamics, the quadcopter state as part of the state variable). The sketch example provided at the beginning of section 4 was only meant to provide some intuition around the potential consequences of confounding in the imitation learning setting.
> 6. We appreciate the question about the connection between PRMSE and recovering $E[Y|do(X)]$ as we should have worded it better. We have updated our draft with what we hope is more clear wording. In short, we are in fact estimating the effect of interventions: a low PRMSE is a necessary condition for recovering $E[Y|do(X)]$. For it to be a sufficient condition, one needs the natural identifiability assumptions (e.g. full rank for the linear setting) and for the instrument to be informative about X -- Chen and Pouzo discuss this point in far greater detail.
> 7. We hope the above points are evidence that we are indeed attempting to compute a causal effect and not merely doing standard regression.

---

### Decision · Program_Chairs · 2022-01-20

**Decision:**

Reject

**Comment:**

This paper studies imitation learning from a causal inference perspective. The authors propose a method to remove the effects of confounders on expert action a using instrumental variable regression, which presumably leads to better estimation of P(a|s), and hence better imitation.  The reviews were negative overall at the start. After the discussions, one reviewer stated that he would change his recommendation to accept, although his score is not changed on the review form. However, another reviewer is still not convinced that the causal formalism introduced in the paper improves over the existing RL literature.